# Approximate inference of marginals using the IBIA framework

**Shivani Bathla**
Department of Electrical Engineering
Indian Institute of Technology Madras
India, 600036
ee13s064@ee.iitm.ac.in

**Vinita Vasudevan**
Department of Electrical Engineering
Indian Institute of Technology Madras
India, 600036
vinita@ee.iitm.ac.in

## Abstract

Exact inference of marginals in probabilistic graphical models (PGM) is known to be intractable, necessitating the use of approximate methods. Most of the existing variational techniques perform iterative message passing in loopy graphs which is slow to converge for many benchmarks. In this paper, we propose a new algorithm for marginal inference that is based on the incremental build-infer-approximate (IBIA) paradigm. Our algorithm converts the PGM into a *sequence of linked clique tree forests (SLCTF)* with bounded clique sizes, and then uses a heuristic belief update algorithm to infer the marginals. For the special case of Bayesian networks, we show that if the incremental build step in IBIA uses the topological order of variables then (a) the prior marginals are consistent in all CTFs in the SLCTF and (b) the posterior marginals are consistent once all evidence variables are added to the SLCTF. In our approach, the belief propagation step is non-iterative and the accuracy-complexity trade-off is controlled using user-defined clique size bounds. Results for several benchmark sets from recent UAI competitions show that our method gives either better or comparable accuracy than existing variational and sampling based methods, with smaller runtimes.

## 1 Introduction

Discrete probabilistic graphical models (PGM) including Bayesian networks (BN) and Markov networks (MN) are used for probabilistic inference in a wide variety of applications. An important task in probabilistic reasoning is the computation of posterior marginals of all the variables in the network. Exact inference is known to be #P-complete [Roth, 1996], thus necessitating approximations. Approximate techniques can be broadly classified as sampling based and variational methods.

Sampling based methods include Markov chain Monte Carlo based techniques like Gibbs sampling [Gelfand, 2000, Kelly et al., 2019] and importance sampling based methods [Gogate and Dechter, 2011, Friedman and Van den Broeck, 2018, Kask et al., 2020, Broka, 2018, Lou et al., 2019, 2017a,b, Marinescu et al., 2019, 2018]. An advantage of these methods is that accuracy can be improved with time without increasing the required memory. However, in many benchmarks the improvement becomes slow with time. Moreover, many of the recent sampling/search based techniques Kask et al. [2020], Broka [2018], Lou et al. [2019, 2017a,b], Marinescu et al. [2019, 2018] have been evaluated either for approximate inference of partition function (PR) or for finding the marginal maximum a posteriori assignment (MMAP). Currently, there are no published results for posterior marginals (MAR) using these methods, and the publicly available implementations do not support the MAR task. Alternatively, variational techniques can be used. These include loopy belief propagation (LBP) [Frey and MacKay, 1998] region-graph based techniques like generalized belief propagation (GBP) [Yedidia et al., 2000] and its variants [Heskes et al., 2003, Mooij and Kappen, 2007, Lin et al., 2020], mini-bucket based schemes like iterative join graph propagation

(IJGP) [Mateescu et al., 2010] and weighted mini-bucket elimination (WMB) [Liu and Ihler, 2011] and methods that simplify the graph structure like edge deletion belief propagation (EDBP) and the related relax-compensate-recover (RCR) techniques [Choi et al., 2005, Choi and Darwiche, 2006, 2010]. While the accuracy-complexity trade-off can be achieved using a single user-defined clique size bound in mini-bucket based methods, it is non-trivial in many of the other region graph based methods. Most of these techniques use iterative message passing to solve an optimization problem, for which convergence is not guaranteed and even if possible, can be slow to achieve. Non-iterative methods like Deep Bucket Elimination (DBE) [Razeghi et al., 2021] and NeuroBE [Agarwal et al., 2022] are extensions of bucket elimination that approximate messages using neural networks. However, training these networks takes several hours. Moreover, the publicly available implementations of these methods do not support the MAR task.

The recently proposed *incremental build-infer-approximate* (IBIA) framework [Bathla and Vasudevan, 2023] uses a different approach. It converts the PGM into a sequence of calibrated clique tree forests (SCTF) with clique sizes bounded to a user-defined value. Bathla and Vasudevan [2023] show that the normalization constant (NC) of clique beliefs in the last CTF in the sequence is a good approximation of the partition function of the overall distribution. This framework has two main advantages. Firstly, since it is based on clique trees and not loopy graphs, the belief propagation step is non-iterative. Therefore, it is fast and has no issues related to convergence. Secondly, it provides an easy control of the accuracy complexity trade-off using two user-defined parameters and hence can be used in anytime manner. However, the framework in Bathla and Vasudevan [2023] cannot be used to infer marginals. This is because only the clique beliefs in the last CTF account for all factors in the PGM. Beliefs in all other CTFs account for a subset of factors and thus, cannot be used for inference of marginals.

**Contributions of this work:** In this paper, we propose a method for marginal inference that uses the IBIA framework. We show that the approximation algorithm used in this framework preserves the within-clique beliefs. Based on this property, we modify the data structure generated by IBIA to add links between adjacent CTFs. We refer to the modified data structure as a *sequence of linked clique tree forests* (SLCTF). We propose a heuristic belief update algorithm that back-propagates beliefs from the last CTF to the previous CTFs via the links and re-calibrates each CTF so that the updated beliefs account for all factors in the PGM. We also propose a greedy heuristic for the choice of links used for belief update. Results for several UAI benchmark sets show that our method gives an accuracy that is better than or comparable to the existing variational and sampling based methods, with competitive runtimes.

For the special case of BNs, we show that if the incremental build step in IBIA is performed in the topological order of variables then (a) the estimated partition function is guaranteed to be one if no evidence variables are present (b) the prior marginals of all variables are consistent across all CTFs in the sequence and (c) once all the evidence variables have been added to the SLCTF, the posterior marginals of variables in subsequent CTFs are consistent. Our results show that using the topological ordering for BNs leads to better estimates of partition function, prior marginals and posterior marginals in most benchmarks.

## 2 Background

### 2.1 Discrete Probabilistic Graphical Models

Let $\mathcal{X} = \{X_1, X_2, \cdots X_n\}$ be a set of random variables with associated domains $D = \{D_{X_1}, D_{X_2}, \cdots D_{X_n}\}$. The probabilistic graphical model (PGM) over $\mathcal{X}$ consists of a set of factors, $\Phi$, where each factor $\phi_\alpha(\mathcal{X}_\alpha) \in \Phi$ is defined over a subset of variables, $Scope(\phi_\alpha) = \mathcal{X}_\alpha$. If $D_\alpha$ denotes the Cartesian product of the domains of variables in $\mathcal{X}_\alpha$, then $\phi_\alpha : D_\alpha \to R \geq 0$. The joint probability distribution captured by the PGM is $P(\mathcal{X}) = \frac{1}{Z} \prod_\alpha \phi_\alpha$, where $Z = \sum_{\mathcal{X}} \prod_\alpha \phi_\alpha$ is the partition function. PGMs can be broadly classified as Markov networks (MN) which are the undirected models and Bayesian networks (BN) which are the directed models.

One method to perform exact inference involves converting the PGM into a *clique tree* (CT), which is a hypertree where each node $C_i$ is a clique that contains a subset of variables. We use the term $C_i$ as a label for the clique as well as to denote the set of variables in the clique. An edge between $C_i$ and $C_j$ is associated with a set of *sepset variables* denoted $S_{i,j} = C_i \cap C_j$. Exact inference in a CT is done using the belief propagation (BP) algorithm [Lauritzen and Spiegelhalter, 1988] that is

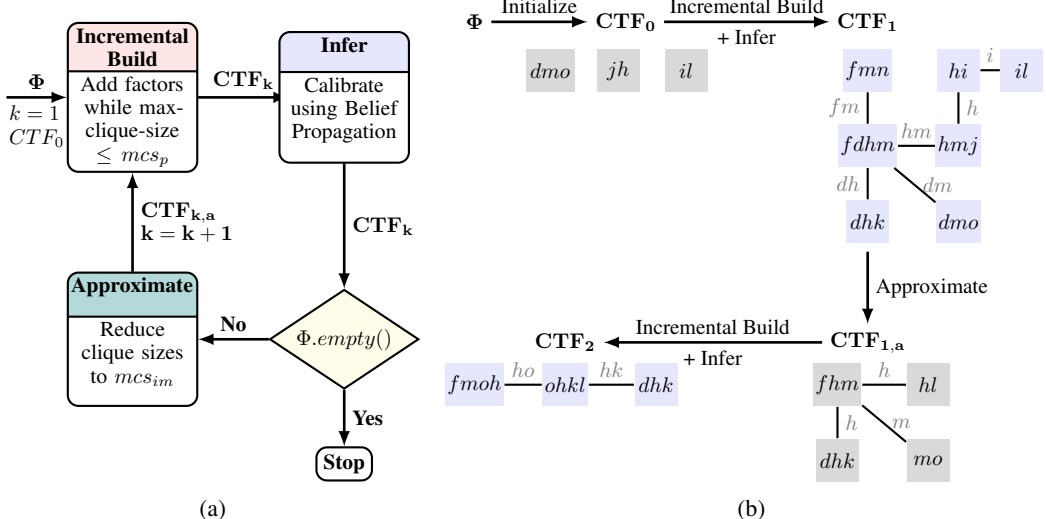

Figure 1: Conversion of the PGM, $\Phi$, into a sequence of calibrated CTFs (SCTF) using the IBIA framework. (a) Overall methodology. (b) Construction of $SCTF = \{CTF_1, CTF_2\}$ for a PGM, $\Phi = \{\phi(d,m,o),\phi(j,h),\phi(i,l),\phi(f,m,n),\phi(h,i),\phi(d,h,k),\phi(f,d,h),\phi(j,m),\phi(k,l,o),\phi(f,o)\}$, with $mcs_p$ and $mcs_{im}$ set to 4 and 3 respectively. $CTF_0$ is formed using cliques corresponding to factors $\phi(d,m,o),\phi(j,h),\phi(i,l)$. $CTF_1$ contains all factors except $\phi(k,l,o),\phi(f,o)$. These factors are added to $CTF_2$.

equivalent to two rounds of message passing along the edges of the CT. The algorithm returns a CT with calibrated clique beliefs $\beta(C_i)$ and sepset beliefs $\mu(S_{i,j})$. In a calibrated CT, all clique beliefs have the same normalization constant ($Z$) and beliefs of all adjacent cliques agree over the marginals of the sepset variables. The joint probability distribution, $P(\mathcal{X})$, can be rewritten as follows.

$$P(\mathcal{X}) = \frac{1}{Z} \frac{\prod_{i \in \mathcal{V}_T} \beta(C_i)}{\prod_{(i,j) \in \mathcal{E}_T} \mu(S_{i,j})} \tag{1}$$

where $\mathcal{V}_T$ and $\mathcal{E}_T$ are the set of nodes and edges in the CT. The marginal probability distribution of a variable $X_i$ can be obtained by marginalizing the belief of any clique $C$ that contains $X_i$ as follows.

$$P(X_i) = \frac{1}{Z} \sum_{C \setminus X_i} \beta(C) \tag{2}$$

We use the following definitions in this paper.

**Definition 1.** Clique Tree Forest (CTF): Set of disjoint clique trees.

**Definition 2.** Clique size: The clique size $cs$ of a clique $C$ is the effective number of binary variables contained in $C$. It is computed as follows.

$$cs = \log_2 \left( \prod_{X_i \in C} |D_{X_i}| \right) \tag{3}$$

where $|D_{X_i}|$ is the cardinality or the number of states in the domain of the variable $X_i$.

**Definition 3.** Prior marginals ($P(X_i)$): It is the marginal probability of a variable $X_i$ when the PGM has no evidence variables.

**Definition 4.** Posterior marginals ($P(X_i|E = e)$): It is the conditional probability distribution of a variable $X_i$, given a fixed evidence state $e$.

## 2.2 Overview of IBIA framework

**Methodology:** The IBIA framework proposed in Bathla and Vasudevan [2023] converts the PGM into a sequence of calibrated CTFs (SCTF) with bounded clique sizes. IBIA starts with an initial

CTF, $CTF_0$, that contains cliques corresponding to factors with disjoint scope. Figure 1a illustrates the overall methodology used in the framework. It uses three main steps as described below.

*Incremental Build:* In this step, a CTF, $CTF_k$, is constructed by incrementally adding factors in the PGM to an initial CTF ($CTF_0$ or $CTF_{k-1,a}$) until the maximum clique size reaches a user-specified bound $mcs_p$. Methods for incremental construction of CTs have been proposed in Bathla and Vasudevan [2023] and Flores et al. [2002]. Either of the two methods can be used to incrementally add new factors.

*Infer:* In this step, all CTs in $CTF_k$ are calibrated using the standard belief propagation algorithm.

*Approximate:* In this step, $CTF_k$ is approximated to give an approximate CTF, $CTF_{k,a}$, that has clique sizes reduced to another user-specified bound $mcs_{im}$.

As shown in Figure 1a, IBIA constructs the $SCTF = \{CTF_1, \ldots, CTF_n\}$ by repeatedly using the incremental build, infer and approximate steps. This process continues until all factors in the PGM are added to some CTF in the SCTF. Figure 1b shows the SCTF generated by IBIA for an example PGM, $\Phi$ (specified in the caption to the figure), with clique size bounds $mcs_p$ and $mcs_{im}$ set to 4 and 3 respectively. For the purpose of illustrating all steps, the disjoint cliques corresponding to factors $\phi(d, m, o), \phi(i, l)$ and $\phi(j, h)$ are chosen as the initial CTF, $CTF_0$. $CTF_1$ is constructed by incrementally adding factors to $CTF_0$. All factors except $\phi(k, l, o)$ and $\phi(f, o)$ are added to $CTF_1$. These two factors are deferred since their addition results in clique sizes greater than $mcs_p = 4$. $CTF_1$ is calibrated using BP and then approximated to give $CTF_{1,a}$ with clique sizes bounded to $mcs_{im} = 3$. $CTF_2$ is constructed by adding the remaining factors to $CTF_{1,a}$. We will use this example to explain the steps in our method for inference of marginals.

**Approximate step:** Since our inference algorithm is based on the properties of the approximate CTF, we explain this step in more detail using the running example shown in Figure 1b. Variables $f, k, l$ and $o$ in $CTF_1$ are also present in the deferred factors $\phi(k, l, o)$ and $\phi(f, o)$. These variables are needed for the construction of subsequent CTFs and are called *interface variables* (IV). All other variables in $CTF_1$ are called *non-interface variables* (NIV). $CTF_{1,a}$ is initialized as the minimal subgraph that connects the IVs. This subgraph contains all cliques in $CTF_1$ except clique $fmn$. Approximation involves two main steps to reduce the number of cliques and clique sizes.
1. *Exact marginalization:* The goal of this step is to remove as many NIVs as possible while ensuring that the overall joint distribution is preserved. NIV $j$ is present in a single clique $hmj$ and is marginalized out from it by summing over the states of $j$. The resulting clique $hm$ is a non-maximal clique that is contained in clique $fdhm$, and is thus removed. NIV $i$ is removed after collapsing the two containing cliques $hi$ and $il$. Exact marginalization of the other NIVs results in collapsed cliques with size greater than $mcs_{im} = 3$, and is not performed.
2. *Local marginalization:* In this step, clique sizes are reduced by marginalizing variables from individual cliques with size greater than $mcs_{im}$ while ensuring that (a) $CTF_{1,a}$ is a valid CTF that satisfies the running intersection property (RIP) (b) a connected CT in $CTF_1$ remains connected in $CTF_{1,a}$ and (c) $CTF_{1,a}$ contains all IVs. To reduce the size of the large-sized clique $fdhm$, NIV $d$ is locally marginalized from this clique. In order to satisfy RIP, it needs to be marginalized from either clique $dmo$ or $dhk$ as well as the corresponding sepsets. Here, $d$ is locally marginalized from clique $dmo$ and the corresponding sepsets to give $CTF_{1,a}$ as shown in the figure. Since all cliques containing $d$ are not collapsed before marginalization, this results in an approximate joint distribution. Propositions 5 and 6 in Bathla and Vasudevan [2023] show that all CTs in the approximate CTF obtained after exact and local marginalization are valid and calibrated.

## 3 Inference of marginals

In this section, we first discuss some of the properties satisfied by the SCTF generated by IBIA. Based on these properties, we then describe the proposed methodology for inference of marginals.

### 3.1 Properties of SCTF

We show that each $CTF_k$ in SCTF and the corresponding approximate CTF, $CTF_{k,a}$, satisfy the following properties. The detailed proofs for these properties are included in the supplementary material and the main ideas used in the proofs are discussed here.

**Proposition 1.** *The joint belief of variables contained within any clique in the approximate CTF, $CTF_{k,a}$, is the same as that in $CTF_k$.*

*Proof Sketch.* Both exact and local marginalization involve summing clique beliefs over the states of a variable, which does not alter the joint belief of the remaining variables in the clique. □

**Proposition 2.** *The clique beliefs in $CTF_k$ account for all factors added to {$CTF_1, \ldots, CTF_k$}.*

*Proof Sketch.* $CTF_1$ is exact, with clique beliefs corresponding to the partial set of factors used to form $CTF_1$. $CTF_{1,a}$ is a calibrated CTF that is obtained after approximating $CTF_1$. Thus, the joint belief of variables in $CTF_{1,a}$ is an approximation of the beliefs encoded by factors added to $CTF_1$. $CTF_2$ is obtained after adding new factors to $CTF_{1,a}$. Therefore, after calibration, clique beliefs in $CTF_2$ account for factors added to $CTF_1$ and $CTF_2$. The proposition follows, since the same argument holds true for all CTFs in the sequence. □

BNs can be handled in a similar manner as MNs by using the undirected moralized graph corresponding to the BN. Another possibility is to use the directed acyclic graph (DAG) corresponding to the BN to guide the incremental build step. In contrast to MNs where all factors are un-normalized, each factor in a BN is the conditional probability distribution (CPD) of a variable $y$ given the state of its parents in the DAG ($Pa_y$). Factors corresponding to the evidence variables are simplified based on the given state and hence become un-normalized.

For BNs, the following properties hold true if each CTF in the SCTF is built by adding factors in the topological order. By this, we mean that the factor corresponding to the variable $y$ is added only after the factors corresponding to all its parent variables, $Pa_y$, have been added to some CTF in the sequence. Let $Y_k$ denote the set of variables whose CPDs are added during construction of $CTF_k$, $e_k$ denote the evidence states of all evidence variables in $Y_k$ and $Pa_{Y_k}$ denote the parents of variables in the set $Y_k$.

**Proposition 3.** *The product of factors added in CTFs, {$CTF_1, \ldots, CTF_k$} is a valid joint probability distribution whose normalization constant is the probability of evidence states $e_1, \ldots, e_k$.*

*Proof.* For each variable $y \in \{Y_1, \ldots, Y_k\}$, the corresponding CPD, $P(y|Pa_y)$, is added to some CTF in {$CTF_1, \ldots, CTF_k$}. The proposition follows since the CPDs corresponding to parents of $y$ are always added to a CTF before the CPD of $y$ is added. □

**Proposition 4.** *The normalization constant of the distribution encoded by the calibrated beliefs in $CTF_k$ is the estimate of probability of evidence states $e_1, \ldots, e_k$.*

*Proof Sketch.* Using Proposition 3, the normalization constant (NC) of the distribution encoded by $CTF_1$ is $P(e_1)$. Using Proposition 1, the approximation algorithm preserves the within-clique beliefs and hence the NC. Thus, the NC of $CTF_{1,a}$ is also $P(e_1)$. Although the NC is the same, the overall distribution corresponding to $CTF_{1,a}$ is approximate due to local marginalization. $CTF_2$ is constructed by adding CPDs of variables in $Y_2$ to $CTF_{1,a}$. CPDs of parent variables in $Pa_{Y_2}$ are added either in $CTF_1$ or $CTF_2$. Hence, after calibration, we get a valid probability distribution with NC as the estimate of probability of evidence states $e_1, e_2$. A similar procedure can be used to show that the property holds for all CTFs. □

**Corollary 1.** *For a BN with no evidence variables, the normalization constant of any CT in $CTF_k$ is guaranteed to be one.*

**Theorem 1.** *Let $I_E$ denote the index of the last CTF in the sequence where the factor corresponding to an evidence variable is added. The posterior marginals of variables present in CTFs {$CTF_k, k \geq I_E$} are preserved and can be computed from any of these CTFs.*

*Proof Sketch.* Once all evidence variables are added, additional CPDs added in each new CTF in {$CTF_k, k > I_E$} correspond to the successors in the BN. Since none of the successors are evidence variables, the corresponding CPDs are normalized and hence cannot change the beliefs of the previous variables. □

**Corollary 2.** *For a BN with no evidence variables, the estimate of prior marginals obtained from any CTF in the sequence is the same.*

## 3.2 Proposed algorithm for inference of marginals

We first explain our approach for estimation of marginals with the help of the example shown in Figure 1b. Following this, we formally describe the main steps in our algorithm.

The SCTF generated by IBIA for the example in Figure 1b contains two CTFs, $CTF_1$ and $CTF_2$. Using Proposition 2, we know that calibrated clique beliefs in $CTF_2$ account for all factors in the PGM, $\Phi$. Therefore, the marginals of all variables present in it can be inferred using Equation 2. However, clique beliefs in $CTF_1$ do not account for factors $\phi(k, l, o)$ and $\phi(f, o)$ which were added during the construction of $CTF_2$. Therefore, in order to infer the marginals of variables $n, j, i$ that are present only in $CTF_1$, we need to update the beliefs to account for these two factors.

Figure 2 shows $CTF_1$, $CTF_{1,a}$ and $CTF_2$ for the example. Using Proposition 1, we know that the joint belief of variables present within any clique in $CTF_{1,a}$ is the same as that in $CTF_1$. However, this belief changes when new factors are added during the construction of $CTF_2$. For instance, $\beta(C_2') = \sum_d \beta(C_2) \neq \sum_o \beta(\tilde{C}_2)$. To reflect the effect of new factors added in $CTF_2$, the joint belief of variables in clique $C_2$ can be updated as follows.

$$\beta_{updated}(C_2) = \frac{\beta(C_2)}{\sum_d \beta(C_2)} \sum_o \beta(\tilde{C}_2)$$

To make sure that $CTF_1$ remains calibrated, this must be followed by a single round of message passing in $CTF_1$ with $C_2$ as the root node. It is clear that a similar belief update is needed for all the variables in $CTF_{1,a}$. However, every update and subsequent round of message passing will override the previous updates. Hence, the beliefs in $CTF_1$ will only approximately reflect the effect of additional factors in $CTF_2$. To improve accuracy, we propose a heuristic procedure for belief update sequence.

Formally, the steps in our algorithm are as follows. Variables present in $CTF_{k,a}$ are present in both $CTF_k$ and $CTF_{k+1}$. We refer to these variables as the *link variables*. We first find links between corresponding cliques $C \in CTF_k$, $C' \in CTF_{k,a}$ and $\tilde{C} \in CTF_{k+1}$. Each link $(C, C', \tilde{C})$ is associated with a set of link variables $V_l = C \cap C'$. For the example, links between $CTF_1$, $CTF_{1,a}$ and $CTF_2$, and the corresponding link variables are shown in magenta in Figure 2. The first part of a link contains cliques $C'$ and $C$. It is obtained as follows.

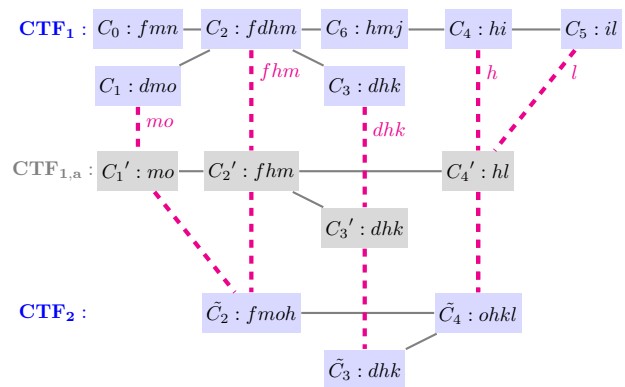

Figure 2: Links between corresponding cliques in $CTF_1$, $CTF_{1,a}$ and $CTF_2$ for the example shown in Figure 1b. All links $(C, C', \tilde{C})$ are marked with dashed magenta lines and the link variables corresponding to each link are marked in magenta color.

(a) If $C'$ is obtained after collapsing a set of cliques $\{C_1, \cdots C_m\}$ in $CTF_k$, $C'$ is linked to each of $\{C_1, \cdots C_m\}$. For example, $C_4'$ is linked to $C_4$ and $C_5$, which were collapsed during exact marginalization of variable $i$.

(b) If $C'$ is obtained from $C$ in $CTF_k$ after local marginalization, $C'$ is linked to $C$. In the example, cliques $C_1'$ and $C_2'$ are obtained after local marginalization of variable $d$ from cliques $C_1$ and $C_2$ respectively. Hence, the corresponding tuples are $(C_1, C_1')$ and $(C_2, C_2')$.

(c) If $C'$ is same as clique $C$ in $CTF_k$, $C'$ is linked to $C$. For example, $C_3$ is linked to $C_3'$.

The second part links $C'$ to $\tilde{C}$ in $CTF_{k+1}$ such that $C' \subseteq \tilde{C}$. This is always possible since $CTF_{k+1}$ is obtained after incrementally modifying $CTF_{k,a}$ to add new factors. Thus, each clique in $CTF_{k,a}$ is contained in some clique in $CTF_{k+1}$. For example, $C_1' \subset \tilde{C}_2$ and the link is $(C_1, C_1', \tilde{C}_2)$.

Let $L_k$ denote the set of links between $CTF_k$ and $CTF_{k+1}$. We refer to the modified data structure consisting of the sequence of calibrated CTFs, $SCTF = \{CTF_k\}$ and a list of links between all adjacent CTFs, $SL = \{L_k\}$, as the *sequence of linked CTFs (SLCTF)*. Once the SLCTF is created, starting from the final CTF in the SLCTF, we successively back-propagate beliefs to the preceding CTFs via links between adjacent CTFs. To back-propagate beliefs from $CTF_{k+1}$ to $CTF_k$, we choose a subset of links in $L_k$ based on heuristics, which will be discussed later. Then, for each selected link $(C, C', \tilde{C})$, we update the belief associated with clique $C$ as follows.

$$\beta_{updated}(C) = \left( \frac{\beta(C)}{\sum\limits_{C \backslash V_l} \beta(C)} \right) \sum_{\tilde{C} \backslash V_l} \beta(\tilde{C}) \tag{4}$$

where, $V_l = C \cap C'$. This is followed by one round of message passing from $C$ to all other cliques in the CT containing $C$. Once all CTFs are re-calibrated, we infer the marginal distribution of a variable using Equation 2 from the last CTF in which it is present.

For BNs, if incremental build is performed by adding variables in the topological order, then as shown in Theorem 1, the singleton marginals are consistent in CTFs $\{CTF_{k \geq I_E}\}$, where $I_E$ is the index of the last CTF to which the factor corresponding to an evidence variable is added. Therefore, in this case, back-propagation of beliefs can be performed starting from $CTF_{I_E}$ instead of starting from the last CTF. This reduces the effort required for belief update.

**Heuristics for choice of links**: To ensure that beliefs of all CTs in $CTF_k$ are updated, at least one link must be chosen for each CT. It is also clear that for any CT in $CTF_k$ more than one link may be required since variables that have low correlations in $CTF_k$ could become tightly correlated when new factors are added in $CTF_{k+1}$. However, belief update via all links is expensive since a round of message passing is required for each link. Based on results over many benchmarks, we propose the following greedy heuristic to choose and schedule links for backward belief update.

(a) To minimize the number of selected links, we first choose a subset of link variables for which the difference in posterior marginals in adjacent CTFs, $CTF_k$ and $CTF_{k+1}$, is greater than a threshold. Next, for belief update, we select a minimal set of links that cover these variables.

(b) The updated beliefs depend on the order in which the links are used for update. Based on the difference in marginals, we form a priority queue with the cliques containing link variables that have the lowest change in marginals having the highest priority. This is to make sure that large belief updates do not get over-written by smaller ones. This could happen for example, if two variables, $v_1$ and $v_2$, that are highly correlated in $CTF_k$ become relatively uncorrelated in $CTF_{k+1}$ due to the approximation. Assume that new factors added to $CTF_{k+1}$ affect $v_1$ but not $v_2$. A belief update via the link containing $v_1$ will make sure that its belief is consistent in $CTF_k$ and $CTF_{k+1}$. Later, if we perform a belief update using a link containing $v_2$, the previous larger belief update of $v_1$ will be overwritten by something smaller since the belief of $v_2$ is not very different in the two CTFs.

**Complexity**: Let $N_{CTF}$ be the number of CTFs in the sequence, and $N_l$ be the maximum number of selected links between any two adjacent CTFs. IBIA requires $2N_{CTF}$ rounds of message passing for calibrating CTFs and $(N_{CTF} - 1)N_l$ rounds for belief update. Therefore, the runtime complexity of IBIA is $O(N_{CTF} N_l 2^{mcs_p})$. To perform backpropagation of beliefs, all CTFs need to be stored and the memory complexity is $O(N_{CTF} 2^{mcs_p})$.

## 4 Results

All experiments were carried out on an Intel i9-12900 Linux system running Ubuntu 22.04.

**Error metrics:** For each non-evidence variable $X_i$, we measure error in terms of *Hellinger distance* between the exact marginal distribution $P(X_i)$ and the approximate marginal distribution $Q(X_i)$. It is computed as follows.

$$HD = \frac{1}{\sqrt{2}} \sqrt{\sum_{s \in Domain(X_i)} \{ \sqrt{P(X_i = s)} - \sqrt{Q(X_i = s)} \}^2} \tag{5}$$

We use two metrics namely, the *average Hellinger distance* (denoted as $HD_{avg}$) and the *maximum Hellinger distance* (denoted as $HD_{max}$) over all non-evidence variables in the network.

Table 1: Comparison of average $HD_{avg}$ and average $HD_{max}$ (shown in gray background) obtained using various inference methods with two runtime constraints, 2 min and 20 min. The minimum error obtained for each time limit is highlighted in bold. Entries are marked as '-' where all instances could not be solved within the set time limit. The total number of instances solved by each method is shown in the last row. $ev_a$: average number of evidence variables, $v_a$: average number of variables, $f_a$: average number of factors, $w_a$: average induced width and $dm_a$: average of the maximum variable domain size.

| | Total | Average stats | 2 min | | | | | 20 min | | | | |
|---|---|---|---|---|---|---|---|---|---|---|---|---|
| | #Inst | $(ev_a, v_a, f_a, w_a, dm_a)$ | LBP | WMB | IJGP | ISSwc | IBIA20 | LBP | WMB | IJGP | ISSwc | IBIA23 |
| BN | 97 | (76,637,637,28,10) | - | 0.037 | - | - | **2E-4** | 0.023 | 0.025 | - | - | **6E-5** |
| | | | - | 0.228 | - | - | **9E-3** | 0.230 | 0.170 | - | - | **2E-3** |
| GridBN | 29 | (0,595,595,37,2) | 0.075 | 0.066 | 0.011 | 0.003 | **5E-6** | 0.075 | 0.048 | 0.010 | 0.001 | **2E-7** |
| | | | 0.478 | 0.416 | 0.111 | 0.051 | **7E-4** | 0.478 | 0.381 | 0.094 | 0.015 | **1E-4** |
| Bnlearn | 26 | (0,256,256,7,16) | 0.010 | 0.005 | 0.011 | 0.012 | **7E-5** | 0.010 | **5E-6** | 0.008 | 0.006 | 7E-6 |
| | | | 0.089 | 0.021 | 0.050 | 0.064 | **0.002** | 0.089 | **1E-4** | 0.025 | 0.028 | 2E-4 |
| Pedigree | 24 | (154,853,853,24,5) | 0.075 | 0.018 | 0.035 | 0.033 | **0.009** | 0.075 | 0.015 | 0.033 | 0.021 | **0.008** |
| | | | 0.555 | 0.253 | 0.470 | 0.292 | **0.204** | 0.555 | **0.194** | 0.446 | 0.234 | 0.198 |
| Promedas | 64 | (7,618,618,21,2) | 0.032 | 0.055 | 0.124 | 0.030 | **0.013** | 0.032 | 0.043 | 0.120 | 0.021 | **0.010** |
| | | | 0.168 | 0.295 | 0.504 | 0.139 | **0.086** | 0.168 | 0.245 | 0.487 | 0.096 | **0.072** |
| DBN | 36 | (653,719,14205,29,2) | - | 0.069 | 0.081 | **0.016** | 0.020 | - | 0.018 | 0.060 | **2E-6** | 0.003 |
| | | | - | 0.883 | 0.919 | 0.766 | **0.261** | - | 0.544 | 0.879 | **2E-4** | 0.098 |
| ObjDetect | 79 | (0,60,210,6,16) | 0.022 | **0.001** | 0.004 | 0.018 | 0.002 | 0.022 | 2E-4 | **3E-5** | 0.009 | 4E-4 |
| | | | 0.130 | **0.010** | 0.037 | 0.189 | 0.020 | 0.130 | 0.003 | **3E-4** | 0.061 | 0.006 |
| Grids | 8 | (0,250,728,22,2) | 0.433 | 0.146 | 0.247 | - | **0.088** | 0.433 | 0.089 | 0.123 | 0.056 | **0.002** |
| | | | 0.905 | 0.343 | 0.713 | - | **0.300** | 0.905 | 0.221 | 0.423 | 0.209 | **0.099** |
| CSP | 12 | (0,73,369,12,4) | 0.019 | 0.033 | 0.026 | - | **0.002** | 0.019 | 0.022 | 0.017 | 0.054 | **2E-4** |
| | | | 0.066 | 0.101 | 0.134 | - | **0.011** | 0.066 | 0.057 | 0.073 | 0.093 | **0.003** |
| Segment | 50 | (0,229,851,17,2) | 0.035 | 1E-4 | **5E-6** | **5E-6** | 6E-5 | 0.035 | 5E-6 | 5E-6 | 5E-6 | **1E-7** |
| | | | 0.258 | 0.002 | **7E-5** | **7E-5** | 0.001 | 0.258 | 7E-5 | 7E-5 | 7E-5 | **4E-7** |
| Protein | 68 | (0,59,176,6,77) | **5E-4** | 0.005 | 0.003 | 0.003 | 6E-4 | 5E-4 | 0.003 | 0.007 | 0.001 | **3E-5** |
| | | | **0.007** | 0.102 | 0.094 | 0.049 | 0.039 | 0.007 | 0.066 | 0.230 | 0.015 | **0.002** |
| #Inst | 493 | | 481 | 493 | 487 | 485 | 493 | 485 | 493 | 491 | 487 | 493 |

**Benchmarks:** We used the benchmark sets included in UAI repository [Ihler, 2006] and the Bayesian network repository [Scutari, 2007]. We classify instances for which exact solutions are present in the repository as *'small'* and others as *'large'*.

**Methods used for comparison:** In the UAI 2022 inference competition [UAI, 2022], the $uai14\_mar$ solver had the highest score for the MAR task. It is an amalgam of solvers that dumps solutions with different methods based on the given time and memory constraints. It uses loopy BP (LBP), generalized BP on loopy graphs where outer regions are selected using mini-bucket heuristics, and cutset conditioning of GBP approximations. The implementation of this solver is not publicly available. Therefore, we have compared our results individually with methods that belong to categories of methods used in $uai14\_mar$. This includes LBP [Murphy et al., 1999], IJGP [Mateescu et al., 2010] and sample search [Gogate and Dechter, 2011] which is an importance sampling based technique that uses an IJGP based proposal and cutset sampling (referred to as 'ISSwc' in this paper). We also compare our results with weighted mini-bucket elimination (WMB) [Liu and Ihler, 2011]. Additional results showing a comparison with results published in Kelly et al. [2019] are included in the supplementary material.

**Evaluation setup:** The implementation of LBP and WMB were taken from LibDAI [Mooij, 2010, 2012] and Merlin [Marinescu, 2016] respectively. For IJGP and ISSwc, we used implementations [Gogate, 2010, 2020] provided by the authors of these methods. LBP, IJGP, ISSwc and WMB are implemented in C++. IBIA on the other hand has been implemented in Python3 and is thus, at a disadvantage in terms of runtime. We report results with runtime limits of 2 min and 20 min for small instances. In all cases, the memory limit was set to 8GB, which is the same as that used in UAI 2022 competition [UAI, 2022]. For IBIA, we set the maximum clique size bound $mcs_p$ to 20 (referred to as *'IBIA20'*) when the time limit is 2 min and we set it to 23 (referred to as *'IBIA23'*) when the time limit is 20 min. $mcs_{im}$ is empirically chosen as 5 less than $mcs_p$. The evaluation setup used for other methods is included in the supplementary material.

**Results:** Table 1 has the results for the small benchmark sets. It reports the average of $HD_{avg}$ and $HD_{max}$ over all instances in each set. We compare results obtained using LBP, WMB, IJGP, IBIA20, IBIA23 and ISSwc for both time constraints. The minimum error obtained for each time limit is marked in bold. IBIA20 and IBIA23 solve all small instances within 2 min and 20 min respectively. In 2 min, the accuracy obtained with IBIA20 is better than all other solvers for most benchmarks. For ObjDetect, Segment and Protein, it is comparable to WMB, IJGP/ISSwc and LBP respectively, which give the least errors for these testcases. In 20 min, IBIA23 gives lower or comparable errors in all testcases except DBN and ObjDetect. For DBNs, ISSwc reduces to exact inference in most instances and hence error obtained is close to zero. For ObjDetect, IJGP gives the least error closely followed by IBIA23. *Note that for many benchmarks the accuracy obtained with IBIA20 in 2 min is either better than or comparable to the accuracy obtained with other solvers in 20 min.*

A comparison of IBIA with results published in Kelly et al. [2019] for Gibbs sampling with Rao-blackwellisation (ARB) and IJGP is included in the supplementary material. It is seen that error obtained with IBIA is lower than both methods in majority of the testcases.

For BN instances, Table 2 compares the results obtained using IBIA20 when CTFs are constructed by adding factors in the topological order (columns marked as 'TP') with that obtained using a non-topological order (columns marked as 'NTP'). We compare the maximum error in partition function (PR) and the average $HD_{max}$ over all instances in each benchmark set.

We observe that the topological ordering gives better accuracy for both PR and marginals in all testcases except Pedigree. The advantage of this ordering is that once all the evidence variables are added, no belief-update is needed is needed for the subsequent CTFs (using Theorem 1). So, the number of belief update steps is lower, resulting in lower errors. However, a drawback of this ordering is that it is rigid and it sometimes results in a larger number of CTFs in the sequence which could lead to larger errors if all the evidence variables are added in later CTFs. When no evidence variables are present (e.g.

Table 2: Comparison of maximum error in PR and average $HD_{max}$ obtained using IBIA20 with CTFs constructed by adding factors in topological order (shown in columns marked 'TP') and that obtained using a non-topological order (shown in columns marked 'NTP'). $ev_a$: Average number of evidence variables, $\Delta_{PR} = |\log_{10} PR - \log_{10} PR^*|$ where $PR$ and $PR^*$ are estimated and exact values.

|  | #Inst | $ev_a$ | Max $\Delta_{PR}$ NTP | Max $\Delta_{PR}$ TP | Avg $HD_{max}$ NTP | Avg $HD_{max}$ TP |
|---|---|---|---|---|---|---|
| Bnlearn | 26 | 0 | 0.02 | **0** | 0.023 | **0.002** |
| GridBN | 29 | 0 | 0.09 | **0** | 0.231 | **0.001** |
| Promedas | 64 | 7 | 1.5 | **0.4** | 0.322 | **0.086** |
| BN | 97 | 76 | 0.07 | **0.02** | 0.116 | **0.009** |
| Pedigree | 24 | 159 | **0.4** | 0.7 | **0.098** | 0.204 |

GridBN, Bnlearn), both runtime and memory complexity is lower with topological ordering since marginals are consistent in all CTFs (using Corollary 2) and belief update is not needed. The average runtime with and without topological ordering was 1s and 146s respectively for GridBN instances and 0.3s and 1.3s for Bnlearn testcases.

To evaluate the scalability of the proposed algorithm, we ran it for large networks where the exact solutions are not known. Table 3 tabulates the percentage of large instances in each benchmark set that could be solved using IBIA within 2 min, 20 min and 60 min. For this experiment, we start with $mcs_p=20$ and allow it to increase if incremental build results in a CTF with larger clique sizes. IBIA could solve all *large* instances in benchmark sets BN, Promedas, ObjDetect and Segmentation and most instances in Protein within 20 min. For other benchmarks, additional

Table 3: Percentage of large instances in each benchmark set solved by IBIA within 2, 20 and 60 minutes. $ev_a$: average number of evidence variables, $v_a$: average number of variables, $f_a$: average number of factors, $w_a$: average induced width and $dm_a$: average of the maximum domain-size.

|  | Total #Inst | Average stats $(ev_a, v_a, f_a, w_a, dm_a)$ | Instances solved (%) 2 min | 20 min | 60 min |
|---|---|---|---|---|---|
| BN | 22 | (188,1272,1272,51,17) | 64 | 100 | 100 |
| Promedas | 171 | (15,1207,1207,71,2) | 77 | 100 | 100 |
| ObjDetect | 37 | (0,60,1830,59,17) | 27 | 100 | 100 |
| Segment | 50 | (0,229,851,19,21) | 100 | 100 | 100 |
| Protein | 395 | (0,306,1192,21,81) | 75 | 97 | 98 |
| DBN | 78 | (784,944,47206,60,2) | 38 | 77 | 77 |
| Grids | 19 | (0,3432,10244,117,2) | 16 | 37 | 58 |
| CSP | 54 | (0,294,11725,175,41) | 31 | 54 | 59 |
| Type4b | 82 | (4272,10822,10822,24,5) | 0 | 9 | 29 |

instances could be solved when the runtime was increased to 60 min. The memory required for the remaining Grids, DBN, CSP and Protein instances is more than 8 GB. The increased memory usage is due to the following reasons. Firstly, all calibrated CTFs in the SLCTF need to be stored in order to allow for back-propagation of beliefs and the memory required increases with the number of CTFs. The average number of CTFs in the remaining Grid, DBN and CSP benchmarks is 22, 58 and 80 respectively. Secondly, for benchmarks with large variable domain sizes, the number of variables present in each clique in a CTF is small. Therefore, approximation using exact and local marginalization becomes infeasible and the subsequent CTFs have clique sizes greater than $mcs_p$, which results in increased memory usage. This is seen in 9 out of 395 Protein instances and 12 out of 54 CSP instances. In addition to memory, the remaining Type4b instances also require additional runtime. This is because during belief update of each CTF, we perform one round of message passing for each selected link and the number of links is large in these instances.

## 5   Discussion

**Limitations:** While the belief update algorithm performs well for most benchmarks, it has some limitations. It is sequential and is performed link by link for each CTF that needs to be updated. The time and space complexity depends on the number of CTFs in the sequence and the number of selected links, which is large in some testcases. Also, after belief-update of all CTFs is completed, beliefs of variables present in multiple CTFs need not be consistent. However, good accuracies are obtained when beliefs are inferred from the last CTF containing the variable. For BNs, we found that building CTFs in the topological order gives larger errors in some cases. A possible extension would be to have an efficient build strategy where the ordering is decided dynamically based on the properties of the graph structure.

**Comparison with related work**: IBIA is similar to mini-bucket schemes in the sense that the accuracy-complexity tradeoff is controlled using a user-defined maximum clique size bound. While mini-bucket based schemes like IJGP [Dechter et al., 2002] and join graph linear programming [Ihler et al., 2012] use iterative message passing in loopy graphs, others like mini-bucket elimination (MBE), WMB [Liu and Ihler, 2011] and mini-clustering [Mateescu et al., 2010] are non-iterative approaches that approximate the messages by migrating the sum operator inside the product term. In contrast, IBIA constructs a sequence of clique trees. It performs belief propagation on approximate clique trees so that messages are exact and there are no issues of convergence.

Unlike sampling based techniques, there is no inherent randomness in IBIA that is each run gives the same results. There could be a variation if the order in which factors are added is changed. However, this variation is minimal since subsets of factors are added together in the incremental build step. In that sense, it is like mini-bucket based methods where results are the same if the variable elimination order and the partitioning technique used to generate the initial choice of mini-buckets are the same.

In order to construct the sequence, IBIA requires a fast and accurate method to approximate clique trees by reducing clique sizes. The aim is to preserve as much as possible, the joint distribution of the interface variables. This is achieved by marginalizing out variables from large-sized cliques and a minimal set of neighbors without disconnecting clique trees. IBIA gives good accuracy since variables that are removed are chosen based on a mutual information (MI) based metric and a sufficient number of non-interface variables are retained so that a CT is never disconnected. In contrast, the Boyen Koller (BK) [Boyen and Koller, 1998, Murphy, 2002] and factored frontier (FF) approximations, although fast, retain only the interface variables which can disconnect the CT resulting in larger errors due to the underlying independence approximation. The thin junction tree approximations proposed in Kjærulff [1994] and Hutter et al. [2004] split cliques that contain a pair of variables that is not present in any of its sepsets. However, if there are large cliques that have only sepset variables (which is typical in most benchmarks), then the split has to be done iteratively starting from leaf nodes of multiple branches, until the large-sized clique can be split. When such cliques are centrally located in the CT, this process is both time-consuming and would result in approximation of a much larger set of cliques. Similar to BK and FF, this method can also disconnect the CT.

The other thin junction tree methods [Bach and Jordan, 2001, Elidan and Gould, 2008, Dafna and Guestrin, 2009] choose an optimum set of features based on either KL distance or a MI based metric. They cannot be directly used since IBIA requires a good approximation of the joint distribution of only the interface variables. Also, these methods are typically iterative and not very fast.

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
