# Supplementary material

**Shivani Bathla**
Department of Electrical Engineering
Indian Institute of Technology Madras
India, 600036
ee13s064@ee.iitm.ac.in

**Vinita Vasudevan**
Department of Electrical Engineering
Indian Institute of Technology Madras
India, 600036
vinita@ee.iitm.ac.in

## A  Results

### A.1  Evaluation setup

For loopy belief propagation (LBP) [Murphy et al., 1999], we use the implementation provided in LibDAI [Mooij, 2010, 2012]. We set the tolerance limit to $10^{-3}$ when time limit is 2 min and $10^{-9}$ for 20 min. For iterative join graph propagation (IJGP) [Mateescu et al., 2010], we used the implementation available on the author's webpage [Gogate, 2010]. The maximum cluster size in IJGP is set using the parameter $ibound$. This solver starts with the minimal value of $ibound$ and increases it until the runtime and memory constraints are satisfied. A solution is obtained for each $ibound$. The results reported are those obtained for the largest $ibound$ possible for the given time and memory constraints. For WMB, we used the implementation made available by the authors in the Merlin tool [Marinescu, 2016]. Since this implementation uses a fixed $ibound$ value, we wrote a script to run it in anytime fashion similar to IJGP. We report results obtained with the largest value of $ibound$ possible. For sample search with IJGP-based proposal and cutset sampling (ISSwc) [Gogate and Dechter, 2011], we used the implementation provided by the authors on Github [Gogate, 2020]. For ISSwc, appropriate values of $ibound$ and $w\text{-}cutset$ bound are set by the tool based on the given runtime limit.

### A.2  Additional results

For a fair comparison with IBIA using $mcs_p$ of 20 (referred to as 'IBIA20'), we also obtained the results for ISSwc after fixing both $ibound$ and $w\text{-}cutset$ bound to 20 (referred to as 'ISSwc20'). Table 1 compares the results obtained using IBIA20, ISSwc20 and ISSwc (in which the optimal $ibound$ is determined by the solver). The runtime limit was set to 2 min and 20 min, and the memory limit was set to 8 GB. The error obtained using IBIA20 is either smaller than or comparable to ISSwc20 and ISSwc for both time limits in all testcases except DBN. For DBN, in 2 min, the average $HD_{max}$ obtained with IBIA20 is significantly smaller than both variants of sample search, and the average $HD_{avg}$ obtained with IBIA20 is comparable. However, in 20 min, both variants reduce to exact inference in many DBN instances and the average error obtained is close to zero.

Table 2 compares the maximum Hellinger distance obtained using IBIA ($mcs_p$=15,20) with published results for adaptive Rao Blackwellisation (ARB) and iterative join graph propagation in Kelly et al. [2019]. The minimum error obtained is shown in bold. IBIA with $mcs_p = 20$ gives the least error in all cases. The error obtained with $mcs_p = 15$ is smaller than ARB and IJGP in all testcases except Grids_11, Grids_13 and Promedas_12.

37th Conference on Neural Information Processing Systems (NeurIPS 2023).

Table 1: Comparison of average $HD_{avg}$ and average $HD_{max}$ (shown in gray background) obtained using IBIA with $mcs_p = 20$ (IBIA20), ISSwc with clique size bounds determined by the solver [Gogate, 2020] (ISSwc) and ISSwc with $ibound$ and $w\text{-}cutset$ bound fixed to 20 (ISSwc20). Results are shown for two runtime limits, 2 min and 20 min. Entries are marked with '-' if the solution for all testcases could not be obtained within the given time and memory limits. The minimum error obtained for a benchmark is highlighted in bold. The number of instances solved by each solver is shown in the last row. $ev_a$: average number of evidence variables, $v_a$: average number of variables, $f_a$: average number of factors, $w_a$: average induced width and $dm_a$: average of the maximum variable domain size.

| | Total #Inst | $(ev_a, v_a, f_a, w_a, dm_a)$ | 2 min | | | 20 min | | |
|---|---|---|---|---|---|---|---|---|
| | | | ISSwc | ISSwc20 | IBIA20 | ISSwc | ISSwc20 | IBIA20 |
| BN | 97 | (76,637,637,28,10) | - | 0.037 | **0** | - | 0.033 | **0** |
| | | | - | 0.145 | **0** | - | 0.085 | **0** |
| GridBN | 29 | (0,595,595,37,2) | 0.003 | 0.005 | **0** | 0.001 | 0.005 | **0** |
| | | | 0.051 | 0.065 | **0** | 0.015 | 0.046 | **0** |
| Bnlearn | 26 | (0,256,256,7,16) | 0.012 | 0.036 | **0** | 0.006 | 0.036 | **0** |
| | | | 0.064 | 0.094 | **0.002** | 0.028 | 0.093 | **0.002** |
| Pedigree | 24 | (154,853,853,24,5) | 0.033 | 0.028 | **0.009** | 0.021 | 0.021 | **0.009** |
| | | | 0.292 | 0.245 | **0.204** | 0.234 | **0.195** | 0.204 |
| Promedas | 64 | (7,618,618,21,2) | 0.030 | 0.042 | **0.013** | 0.021 | 0.033 | **0.013** |
| | | | 0.139 | 0.207 | **0.086** | 0.096 | 0.153 | **0.086** |
| DBN | 36 | (653,719,14205,29,2) | 0.016 | **0.011** | 0.020 | **0** | **0** | 0.020 |
| | | | 0.766 | 0.833 | **0.261** | **0** | **0** | 0.261 |
| ObjDetect | 79 | (0,60,210,6,16) | 0.018 | 0.039 | **0.002** | 0.009 | 0.004 | **0.002** |
| | | | 0.189 | 0.233 | **0.020** | 0.061 | 0.021 | **0.020** |
| Grids | 8 | (0,250,728,22,2) | - | - | **0.088** | 0.056 | - | 0.088 |
| | | | - | - | **0.300** | 0.209 | - | 0.300 |
| CSP | 12 | (0,73,369,12,4) | - | - | **0.002** | 0.054 | 0.069 | **0.002** |
| | | | - | - | **0.011** | 0.093 | 0.081 | **0.011** |
| Segment | 50 | (0,229,851,17,2) | **0** | 0.002 | **0** | **0** | **0** | **0** |
| | | | **0** | 0.036 | 0.001 | **0** | **0** | 0.001 |
| Protein | 68 | (0,59,176,6,77) | 0.003 | 0.003 | **0** | 0.001 | 0.001 | **0** |
| | | | 0.049 | **0.030** | 0.039 | 0.015 | **0.011** | 0.039 |
| #Inst | 493 | | 485 | 488 | 493 | 487 | 489 | 493 |

Table 2: Comparison of maximum Hellinger distance ($HD_{max}$) obtained using IBIA with published results for Gibbs sampling with adaptive Rao Blackwellisation (ARB) and iterative join graph propagation in Kelly et al. [2019]. Results obtained with $mcs_p = 15$ and $mcs_p = 20$ are shown in columns marked as IBIA15 and IBIA20 respectively. Runtimes (in seconds) for IBIA15 and IBIA20 are also shown. Estimates for ARB were obtained within 600 seconds[+] [Kelly et al., 2019] and runtime for IJGP is not reported in Kelly et al. [2019]. The minimum error obtained for each benchmark is marked in bold. $w$: induced width, $dm$: maximum domain size

| | $w$ | $dm$ | $HD_{max}$ | | | | Runtime (s) | |
|---|---|---|---|---|---|---|---|---|
| | | | Merlin (IJGP)[*] | ARB[*] | IBIA15 | IBIA20 | IBIA15 | IBIA20 |
| Alchemy_11 | 19 | 2 | 0.777 | 0.062 | 0.004 | **1E-7** | 3.3 | 2.9 |
| CSP_11 | 16 | 4 | 0.513 | 0.274 | 0.100 | **0.034** | 0.5 | 3.4 |
| CSP_12 | 11 | 4 | 0.515 | 0.275 | 0.028 | **6E-7** | 0.1 | 0.1 |
| CSP_13 | 19 | 4 | 0.503 | 0.290 | 0.085 | **0.051** | 0.9 | 2.9 |
| Grids_11 | 21 | 2 | 0.543 | 0.420 | 0.590 | **0.166** | 1.1 | 3.5 |
| Grids_12 | 12 | 2 | 0.645 | 0.432 | **3E-7** | **3E-7** | 0.0 | 0.0 |
| Grids_13 | 21 | 2 | 0.500 | 0.544 | 0.962 | **0.246** | 1.1 | 3.6 |
| Pedigree_11 | 19 | 3 | 0.532 | 0.576 | 0.016 | **5E-7** | 0.5 | 0.1 |
| Pedigree_12 | 19 | 3 | 0.562 | 0.506 | 0.023 | **4E-7** | 0.3 | 0.1 |
| Pedigree_13 | 19 | 3 | 0.577 | 0.611 | 5E-7 | **5E-7** | 0.1 | 0.1 |
| Promedus_11 | 18 | 2 | 1.000 | 0.373 | 0.049 | **5E-7** | 1.4 | 0.5 |
| Promedus_12 | 20 | 2 | 1.000 | 0.358 | 0.657 | **0.242** | 2.8 | 4.1 |
| Promedus_13 | 10 | 2 | 1.000 | 0.432 | **5E-7** | **5E-7** | 0.4 | 0.4 |

[*] The results tabulated in Kelly et al. [2019] report $-\log_2 HD_{max}$. The table above has the corresponding values of $HD_{max}$.
[+] System used: Ubuntu 18.04, with 16GB of RAM, 6 CPUs and 2 hardware threads per CPU [Kelly et al., 2019].

## B  Pseudo-code

Algorithm 1 shows the steps in the proposed algorithm for the inference of marginals. We first convert the PGM into a sequence of linked CTFs ($SLCTF$) that contains a sequence of calibrated CTFs ($SCTF = \{CTF_k\}$) and a list of links between adjacent CTFs ($SL = \{L_k\}$). Functions $BuildCTF$ and $ApproximateCTF$ are used for incremental construction of CTFs and approximation of CTFs respectively. The steps in these functions are explained in detail in Algorithms 1 and 2 in Bathla and Vasudevan [2023]. Links between adjacent CTFs are found using the function $FindLinks$ and belief update in the SLCTF is performed using the function $BeliefUpdate$. Following this, the marginal of a variable $v$ is inferred from clique beliefs in the last CTF that contains $v$ (line 23).

## C  Proofs

**Notations**

| | |
|---|---|
| $\Phi_k$ | Set of factors added to construct $CTF_k$ |
| $X_k$ | Set of all non-evidence variables in $CTF_k$ |
| $X_{k,a}$ | Set of all non-evidence variables in $CTF_{k,a}$ |
| $Y_k$ | Set of variables in $CTF_k$ but not in $CTF_1, \ldots, CTF_{k-1}$ |
| $Pa_{Y_k}$ | Parents of variables in $Y_k$ in the BN |
| $E_k$ | Set of evidence variables in $Y_k$ |
| $e_k$ | Evidence state corresponding to variables in $E_k$ |
| $C$ | A clique in $CTF_k$ |
| $C'$ | A clique in $CTF_{k,a}$ |
| $SP$ | Sepset associated with an edge in $CTF_k$ |
| $SP'$ | Sepset associated with an edge in $CTF_{k,a}$ |
| $\beta(C)$ | Unnormalized clique belief of clique $C$ |
| $\beta_N(C)$ | Normalized clique belief of clique $C$, $\beta_N(C) = \frac{\beta(C)}{\sum\limits_{v \in C} \beta(C)}$ |
| $Z_k$ | Normalization constant of the distribution encoded by calibrated beliefs in $CTF_k$ |
| $Q_k(X_k)$ | Probability distribution corresponding to $CTF_k$ |
| $Q_{k,a}(X_{k,a})$ | Probability distribution corresponding to $CTF_{k,a}$ |

**Propositions related to inference of marginals:** Let $CTF_k$ be a CTF in the SCTF generated by the IBIA framework and $CTF_{k,a}$ be the corresponding approximate CTF.

**Proposition 1.** *The joint belief of variables contained within any clique in the approximate CTF $CTF_{k,a}$ is the same as that in $CTF_k$.*

*Proof.* The approximation algorithm has two steps, exact marginalization and local marginalization. Exact marginalization involves finding the joint belief by collapsing all cliques containing a variable and then marginalizing the belief by summing over the states of the variable. This does not change the belief of the remaining variables. Local marginalization involves marginalizing a variable from individual cliques and sepsets by summing over its states. Let $C'$ denote the clique obtained after local marginalization of variable $v$ from clique $C$. The updated clique belief ($\beta(C')$) is computed as shown below.

$$\beta(C') = \sum_v \beta(C)$$

Once again, summing over the states of a variable does not alter the joint belief of the remaining variables in the clique.

$\square$

**Algorithm 1** InferMarginals $(\Phi, mcs_p, mcs_{im})$

---

**Input:** $\Phi$: Set of factors in the PGM
  $mcs_p$: Maximum clique size bound for each CTF in the sequence
  $mcs_{im}$: Maximum clique size bound for the approximate CTF
**Output:** $MAR$: Map containing marginals $< variable : margProb >$

1: Initialize: $MAR = <>$       ▷ Map $< variable : margProb >$
  $S_v = \cup_{\phi \in \Phi} Scope(\phi)$       ▷ Set of all variables in the PGM
  $SCTF = [\,]$       ▷ Sequence of calibrated CTFs
  $SL = [\,]$       ▷ List of list of links between all adjacent CTFs
  $k = 1$       ▷ Index of CTF in $SCTF$
2: **while** $\Phi.isNotEmpty()$ **do**       ▷ Convert PGM $\Phi$ to $SLCTF = \{SCTF, SL\}$
3:     **if** $k == 1$ **then**
4:        $CTF_0 \leftarrow$ Disjoint cliques corresponding to factors in $\Phi$ with disjoint scopes
5:        ▷ Add factors to $CTF_0$ using BuildCTF (Algorithm 1 in Bathla and Vasudevan [2023])
6:        $CTF_1, \Phi_1 \leftarrow$ BuildCTF $(CTF_0, \Phi, mcs_p)$    ▷ $\Phi_1$: Subset of factors in $\Phi$ added to $CTF_1$
7:        $\Phi \leftarrow \Phi \setminus \Phi_1$       ▷ Remove factors added to $CTF_1$ from $\Phi$
8:     **else**
9:        ▷ Add factors to $CTF_{k-1,a}$ using BuildCTF (Algorithm 1 in Bathla and Vasudevan [2023])
10:       $CTF_k, \Phi_k \leftarrow$ BuildCTF $(CTF_{k-1,a}, \Phi, mcs_p)$   ▷ $\Phi_k$: Subset of factors in $\Phi$ added to $CTF_k$
11:       $\Phi \leftarrow \Phi \setminus \Phi_k$       ▷ Remove factors added to $CTF_k$ from $\Phi$
12:       $L_{k-1} \leftarrow$ FindLinks $(CTF_{k-1}, CTF_{k-1,a}, CTF_k)$ ▷ $L_{k-1}$: List of links between $CTF_{k-1}, CTF_k$
13:       $SL.append(L_{k-1})$       ▷ Add $L_{k-1}$ to the sequence of links $SL$
14:     **end if**
15:     Calibrate $CTF_k$ using belief propagation
16:     $SCTF.append(CTF_k)$       ▷ Add $CTF_k$ to the sequence $SCTF$
17:     ▷ Reduce clique sizes to $mcs_{im}$ using ApproximateCTF (Algorithm 2 in Bathla and Vasudevan [2023])
18:     $CTF_{k,a} \leftarrow$ ApproximateCTF $(CTF_k, \Phi, mcs_{im})$
19:     $k \leftarrow k + 1$
20: **end while**
21: $SLCTF = \{SCTF, SL\}$       ▷ Sequence of linked CTFs
22: BeliefUpdate$(SLCTF)$       ▷ Re-calibrate CTFs so that beliefs in all CTFs account for all factors
23: $MAR[v] \leftarrow$ Find marginal of $v$ from $CTF_j$ s.t. $v \in CTF_k, v \notin CTF_{k+1}$    $\forall v \in S_v$ ▷ Infer marginals
24:
25: **procedure** FINDLINKS$(CTF_{k-1}, CTF_{k-1,a}, CTF_k)$
26:     ▷ Each link is a triplet consisting of $C \in CTF_{k-1}$, $C' \in CTF_{k-1,a}$ and $\tilde{C} \in CTF_k$
27:     **for** $C' \in CTF_{k-1,a}$ **do**       ▷ Find links corresponding to each clique $C'$ in $CTF_{k-1,a}$
28:        ▷ Find list of corresponding cliques in $CTF_{k-1}$, $L_c$
29:        **if** $C'.isCollapsedClique$ **then**       ▷ $C'$ is obtained after exact marginalization
30:          $L_c \leftarrow$ List of cliques in $CTF_{k-1}$ that were collapsed to form $C'$
31:        **else**       ▷ $C'$ is either obtained after local marginalization or it is present as is in $CTF_k$
32:          $C \leftarrow$ Clique in $CTF_{k-1}$ s.t. $C' \subseteq C$; $L_c = [C]$
33:        **end if**
34:        Find clique $\tilde{C}$ in $CTF_k$ s.t. $C' \subseteq \tilde{C}$
35:        ▷ Add all links corresponding to $C'$
36:        **for** $C \in L_c$ **do** $L_{k-1}.append((C, C', \tilde{C}))$ **end for**
37:     **end for**
38:     **return** $L_{k-1}$
39: **end procedure**
40:
41: **procedure** BELIEFUPDATE$(SLCTF)$
42:     $SCTF, SL = SLCTF$
43:     **for** $k \in len(SCTF)$ down to 2 **do**       ▷ Update beliefs in $\{CTF_k, k < len(SCTF)\}$
44:        $CTF_{k-1} \leftarrow SCTF[k-1]$; $CTF_k = SCTF[k]$; $L_{k-1} = SL[k-1]$
45:        $L_s \leftarrow$ Priority queue with subset of links in $L_{k-1}$ chosen using heuristics described in Section 3.2
46:        **for** $(C, C', \tilde{C}) \in L_s$ **do**    ▷ Back-propagate beliefs from $CTF_k$ to $CTF_{k-1}$ via all selected links
47:          $\beta(C) = \frac{\beta(C)}{\sum_{C' \setminus \{C \cap C'\}} \beta(C)} \sum_{\tilde{C} \setminus \{C \cap C'\}} \beta(\tilde{C})$ ▷ Update $\beta(C) \in CTF_{k-1}$ based on $\beta(\tilde{C}) \in CTF_k$
48:          Update belief of all other cliques in $CTF_{k-1}$ using single pass message passing with $C$ as root
49:        **end for**
50:     **end for**
51: **end procedure**

**Proposition 2.** *The clique beliefs in $CTF_k$ account for all factors added to {$CTF_1, \ldots, CTF_k$}.*

*Proof.* $CTF_1$ is constructed by adding factors to an initial CTF that contains a set of disjoint cliques corresponding to a subset of factors with disjoint scopes. Let $\Phi_1$ be the set of all factors present in $CTF_1$ and $Z_1$ be the corresponding normalization constant. After calibration, the normalized clique belief ($\beta_N(C)$) of any clique $C$ in $CTF_1$ can be computed as follows.

$$\beta_N(C) = \frac{1}{Z_1} \sum_{X_1 \setminus C} \frac{\prod_{C_i \in CTF_1} \beta(C_i)}{\prod_{SP \in CTF_1} \mu(SP)} = \frac{1}{Z_1} \sum_{X_1 \setminus C} \prod_{\phi \in \Phi_1} \phi$$

Therefore, clique beliefs in $CTF_1$ account for all factors in $\Phi_1$.

$CTF_{1,a}$ is a calibrated CTF (refer Proposition 6, Bathla and Vasudevan [2023]) that is obtained after approximate marginalization of the variables in $X_1 \setminus X_{1,a}$. Therefore, the joint distribution of variables in $CTF_{1,a}$ also accounts for all factors in $\Phi_1$. $CTF_2$ is constructed by adding factors in $\Phi_2$ to $CTF_{1,a}$. Therefore, after calibration, the normalized clique belief ($\beta_N(C)$) of any clique $C$ in $CTF_2$ can be computed as follows.

$$\beta_N(C) = \frac{1}{Z_2} \sum_{X_2 \setminus C} \frac{\prod_{C' \in CTF_{1,a}} \beta(C')}{\prod_{SP' \in CTF_{1,a}} \mu(SP')} \prod_{\phi \in \Phi_2} \phi \qquad (1)$$

where, $Z_2$ is the normalization constant of the distribution in $CTF_2$. Using equation 1, the clique beliefs in $CTF_2$ accounts for all factors in $\Phi_1$ and $\Phi_2$.

A similar procedure can be repeated for subsequent CTFs to show that the proposition holds true for all CTFs in the sequence. $\qquad \square$

### Propositions related inference in BNs:

The following propositions hold true for Bayesian networks when **each CTF in the SCTF is constructed by adding factors or conditional probability distributions (CPD) of variables in the topological order**. $Y_k$ denotes the set of variables whose CPDs are added during construction of $CTF_k$ and $e_k$ denotes the evidence states of all evidence variables in $Y_k$.

**Proposition 3.** *The product of factors added in CTFs, {$CTF_1, \ldots, CTF_k$} is a valid joint probability distribution whose normalization constant is the probability of evidence states $e_1, \ldots, e_k$.*

*Proof.* Let $\mathcal{Y}_k = \{Y_1, \ldots, Y_k\}$ and $\varepsilon_k = \{e_1, \ldots, e_k\}$. Since CTFs are constructed by adding CPDs of variables in the topological order, the CPDs of parents $Pa_{\mathcal{Y}_k}$ are present in {$CTF_1, \ldots, CTF_k$}. Therefore, the product of the CPDs is the unnormalized joint probability distribution $P(\mathcal{Y}_k, \varepsilon_k)$. Since the CPDs of all non-evidence variables are normalized to one, the normalization constant is $P(\varepsilon_k)$. $\qquad \square$

**Proposition 4.** *The normalization constant of the distribution encoded by the calibrated beliefs in $CTF_k$ is the estimate of probability of evidence states $e_1, \ldots, e_k$.*

*Proof.* The initial factors assigned to $CTF_1$ are CPDs of variables in $Y_1$. Therefore, using Proposition 3, the NC obtained after calibration is $Z_1 = P(e_1)$.

$CTF_{1,a}$ is obtained after approximation of $CTF_1$. All CTs in $CTF_{1,a}$ are calibrated CTs and the normalization constant of the distribution in $CTF_{1,a}$ is same as that of $CTF_1$ (refer Propositions 6 and 9 in Bathla and Vasudevan [2023]. However, due to local marginalization, the overall distribution represented by $CTF_{1,a}$ is approximate. The probability distribution corresponding to $CTF_{1,a}$ can be written as follows.

$$Q_{1,a}(X_{1,a}|e_1) = \frac{1}{Z_1} \frac{\prod_{C' \in CTF_{1,a}} \beta(C')}{\prod_{SP' \in CTF_{k,a}} \mu(SP')}$$

$$\implies Z_1 Q_{1,a}(X_{1,a}|e_1) = Q_{1,a}(X_{1,a}, e_1) \qquad (2)$$

where $X_{1,a}$ is the set of variables in $CTF_{1,a}$.

$CTF_2$ is obtained after adding a new set of CPDs of variables in $Y_2$ to $CTF_{1,a}$. Let $X_2 = X_{1,a} \cup \{Y_2 \setminus E_2\}$ denote the set of non-evidence variables in $CTF_2$ and $Pa_{Y_2}$ denote the parents of variables in $Y_2$. The NC of the distribution encoded by $CTF_2$ ($Z_2$) can be computed as follows.

$$Z_2 = \sum_{X_2} \frac{\prod_{C' \in CTF_{1,a}} \beta(C')}{\prod_{SP' \in CTF_{1,a}} \mu(SP')} \prod_{y \in Y_2} P(y|Pa_y)$$

$$= \sum_{X_2} Q_{1,a}(X_{1,a}, e_1) P(Y_2, e_2 \mid Pa_{Y_2}) \quad \text{(using Equation 2)} \tag{3}$$

where $e_2$ are evidence states in $Y_2$. Since $X_2 = X_{1,a} \cup \{Y_2 \setminus E_2\}$ and parent variables in $Pa_{Y_2}$ are present either in $X_{1,a}$ or $Y_2$, the above equation can be re-written as follows.

$$Z_2 = \sum_{X_2} Q_2(X_2, e_1, e_2) = Q(e_1, e_2)$$

Therefore, the NC of $CTF_2$ is an estimate of probability of evidence states $e_1$ and $e_2$.

A similar procedure can be repeated for subsequent CTFs to show that the property holds true for all CTFs in the sequence. $\qquad\square$

**Theorem 1.** *Let $I_E$ denote the index of the last CTF in the sequence where the factor corresponding to an evidence variable is added. The posterior marginals of variables present in CTFs $\{CTF_k, k \geq I_E\}$ are preserved and can be computed from any of these CTFs.*

*Proof.* Let $\varepsilon_{I_E} = \{e_1, \ldots, e_{I_E}\}$ be the set of all evidence states. Let $v$ be a variable present in cliques $C_v \in CTF_{I_E}$, $C'_v \in CTF_{I_E,a}$ and $\tilde{C}_v \in CTF_{I_E+1}$ and let $\beta_N(C_v)$, $\beta_N(C'_v)$ and $\beta_N(\tilde{C}_v)$ be the corresponding normalized clique beliefs. From Proposition 1, the unnormalized belief of variable $v$ in $C_v$ is same as that in $C'_v$. Therefore, the normalized posterior marginal of $v$ obtained from $C_v$ (denoted as $Q_{I_E}(v|\varepsilon_{I_E})$)) is the same as that obtained from $C'_v$, as given below.

$$Q_{I_E}(v|\varepsilon_{I_E}) = \sum_{C_v \setminus v} \beta_N(C_v) = \sum_{C'_v \setminus v} \beta_N(C'_v) \tag{4}$$

Since $CTF_{I_E,a}$ is calibrated (Proposition 6 in Bathla and Vasudevan [2023]) and $CTF_{I_E+1}$ is obtained by adding CPDs of variables in $Y_{I_E+1}$ to $CTF_{I_E,a}$, the NC of $CTF_{I_E+1}$ can be computed by summing over all non-evidence variables as follows.

$$Z_{I_E+1} = \sum_{X_{I_E,a}} \frac{\prod_{C' \in CTF_{I_E,a}} \beta(C')}{\prod_{SP' \in CTF_{I_E,a}} \mu(SP')} \sum_{Y_{I_E+1} \setminus E_{I_E+1}} P(Y_{I_E+1}, e_{I_E+1}|Pa_{Y_{I_E+1}})$$

$$= \sum_{X_{I_E,a}} \frac{\prod_{C' \in CTF_{I_E,a}} \beta(C')}{\prod_{SP' \in CTF_{I_E,a}} \mu(SP')} \quad (\because E_{I_E+1} = \varnothing, \sum_{Y_{I_E+1}} P(Y_{I_E+1} \mid Pa_{Y_{I_E+1}}) = 1)$$

$$= Z_{I_E} \quad \text{(using Proposition 9 in Bathla and Vasudevan [2023])}$$

Therefore, the posterior marginal of $v$ in $CTF_{I_E+1}$ (denoted as $Q_{I_E+1}(v|\varepsilon_{I_E})$) can be computed from the clique belief of $\tilde{C}_v$ as follows.

$$Q_{I_E+1}(v|\varepsilon_{I_E}) = \sum_{\tilde{C}_v \setminus v} \beta_N(\tilde{C}_v)$$

$$= \sum_{X_{I_E,a} \setminus v} \frac{1}{Z_{I_E}} \frac{\prod_{C' \in CTF_{I_E,a}} \beta(C')}{\prod_{SP' \in CTF_{I_E,a}} \mu(SP')} \sum_{Y_{I_E+1} \setminus E_{I_E+1}} P(Y_{I_E+1}, e_{I_E+1} \mid Pa_{Y_{I_E+1}})$$

$$= \sum_{C'_v \setminus v} \beta_N(C'_v) \quad (\because C'_v \in CTF_{I_E,a} \text{ and } E_{I_E+1} = \varnothing)$$

$$= Q_{I_E}(v|\varepsilon_{I_E}) \quad \text{(using Equation 4)}$$

The above procedure can be repeated to show that the posterior marginal of $v$ is also consistent in all subsequent CTFs that contain $v$. $\qquad\square$