# OpenReview forum: "Approximate inference of marginals using the IBIA framework"
_NeurIPS.cc/2023/Conference — NeurIPS 2023 poster_

### Official Review · Reviewer_QKYy · 2023-07-04

**Soundness:** 3 good
**Presentation:** 3 good
**Contribution:** 2 fair
**Rating:** 4
**Confidence:** 5

**Summary:**

The paper presents a new approximation scheme for computing posterior marginals in graphical models such as Bayesian networks and Markov networks. The proposed approach is based on a previously introduced scheme called IBIA (Incremental Build Infer Approximate) which was initially developed for approximating the partition function. The idea is to construct a sequence of bounded size clique trees (called a clique tree forest) that eventually spans all the input factors and subsequently compute the posterior marginals by carefully selecting the appropriate cliques from the forest. The experimental evaluation is carried out on standard benchmarks for graphical models and the proposed scheme is compared with several state-of-the-art approximation scheme based on message passing. The results demonstrate that the proposed approximation scheme is competitive and some times outperforms existing approaches.


**Strengths:**

The paper is relatively well written and organised. It is also relatively self contained and most of the concepts are discussed in sufficient detail. The empirical evaluation is sound and the results are presented in a relatively clear manner. Some theoretical properties of the proposed method are also discussed.


**Weaknesses:**

Firstly, the contribution presented in this paper appears to be quite incremental over the previous work the IBIA framework. Namely it consists of the message passing scheme over the additional links between adjacent clique trees. The proposed heuristic for choosing links appears to be quite arbitrary.

Secondly, I think the proposed method is highly sensitive to the order in which factors are added to the clique tree forest, as well as the initial collection of factors chosen for initialisation. Therefore, the paper should discuss this aspect in more detail.

Thirdly, the paper does not mention anything about the convergence of the scheme nor its time/space complexity.


**Questions:**

1. As presented, the performance measure considered assumes that the exact marginals are available. However, many of the problem instances from the experimental section have large induced widths (e.g., 37 for GridBNs) and therefore are difficult to solve exactly, unless a lot of evidence is asserted in the network. Therefore, how many evidence variables are considered for each of the benchmark classes? And also, how do you compute the score when exact inference results are not available?

2. What's the reason for defining the clique size in terms of log instead of just the number of variables in the clique? I found the log based definition confusing.

3. Why do we need to consider two bounds on the clique size? I think the scheme is well defined with only one.


**Limitations:**

The proposed scheme is clearly related to several existing scheme that aim to bound the size of the join-tree. Prominent examples are the mini-bucket scheme for variable elimination, the mini-clustering approach for cluster tree (join tree) elimination, as well as the iterative join-graph propagation scheme (all from Rina Dechter's group). Similarly, the Incremental Thin Junction Tree scheme by Frank Hutter is also looking at bounding the clique size of a join tree by splitting a larger clique into smaller cliques. Therefore, I think a deeper discussion on the connection between these previous schemes and the proposed IBIA based one is clearly called for (it is currently missing from the paper).

The empirical evaluation only considers a subset of the existing message passing schemes for approximating marginals e.g, LBP, IJGP, and SampleSearch. These methods are more than one decade old (if not more), do not really fall into the more recent category of variational inference methods. I think it would be interesting to compare with more recent methods based on weighted mini-buckets and cost-shifting such as those developed in Alex Ihler's group (including the importance sampling based ones). If you look at the recent UAI-2022 competition results, it looks like these kinds of methods performed best in the MAP (and to some extent in the PR category as well).

---

> ### Author Rebuttal · Authors · 2023-08-09
>
> Thank you for your comments and questions. Following are our responses to each question.
>
> >  how many evidence variables in each benchmark class? how do you compute the score when exact results are not available?
>
> We used evidence sets given in the repository and an average can be added to Table 1. Exact results are either available in UAI repository or could be computed using ACE (performs exact inference using weighted model counting). For other benchmarks, we have only reported the number of instances that are solved by IBIA for various time limits.
>
> > What's the reason for defining the clique size in terms of log instead of number of variables?
>
>  We have used $log_2$ because it gives clique sizes as the effective number of binary variables in a clique. The complexity of inference depends on the size of the clique belief table which is directly $O(2^{mcs_p})$. Therefore, it is easy to control the complexity.  Whereas, if we specify the limit as number of variables, and variables have different domain sizes, it could run out of memory.
>
> > Why are there two bounds on clique size?
>
> The second parameter $mcs_{im}$ controls the extent of approximation in the approximate step (see Figure 1, Page 3). It is possible to make this an internal parameter.
>
> > paper does not mention anything about convergence of scheme nor its time/space complexity.
>
>  Our method uses a sequence of clique *trees*, therefore are no issues related to convergence of BP.
>
> Complexity analysis for sequence construction is included in Bathla \& Vasudevan (2023). For the estimation of marginals, additional time is needed for performing belief-update via the selected links which is  $O(N_{CTF}N_l 2^{mcs_p})$ where, $N_{CTF}$ is sequence length and $N_l$ is the maximum number of selected links between adjacent CTFs. In terms of memory, all CTFs need to be stored to allow for backward belief update, $O(N_{CTF}2^{mcs_p})$.
>
> We can add this discussion to our paper.
>
> > methods are more than one decade old (if not more), do not really fall into the more recent category of variational inference methods. it would be interesting to compare with more recent methods based on weighted mini-buckets and cost-shifting such as those developed in Alex Ihler's group (including importance sampling based ones).
>
> We have included a comparison with published results for a recent Gibbs sampling-based technique that uses adaptive Rao-Blackwellisation (ARB) (Kelly, 2019) in the supplementary material (see Table 2, Page 2).
>
> While WMB itself can be used to find MAR (see Table 1 in the pdf attached with the rebuttal), publicly available implementations for extensions such as the cost-shifting scheme JGLP in Merlin, dynamic importance sampling \& abstraction sampling do not support the MAR task. This is also true for solvers submitted for the MAP and MMAP tasks in UAI2022 like toulbar, daoopt and braobb. It does not seem trivial to extend them to support MAR. If the reviewer can point us to any implementation of other solvers that can find MAR, we would be happy to include it in our comparisons.
>
>
> > proposed heuristic for choosing links appears to be quite arbitrary.
>
>  As discussed in Section 3 (Page 7),  the idea is to reduce the number of times BP is performed since it is expensive. The heuristic is not arbitrary, rather it is a greedy approach where we first choose a subset of common variables whose posterior marginals are significantly different in successive CTFs. Next, for belief-update, we select a minimal set of links that cover these variables.
>
> > I think proposed method is highly sensitive to order in which factors are added, as well as initial collection of factors chosen
>
> IBIA uses the following approach:
>
> 1. As explained in Bathla \& Vasudevan (2023), factors are not added one at a time in IBIA. Rather, we group factors that impact overlapping portions of the existing clique tree and add them together. If the addition of a group is not possible, we choose the largest subset that can be added.
>
> 2. The set of disjoint factors used for Bayesian networks is fixed since a topological order is followed. For Markov networks, the number of factors is first reduced by multiplying the factors that are contained in a maximal clique of the MN. Next, factors with large scopes are prioritized while choosing the initial set of disjoint factors.
>
> We want to emphasize that there is no inherent randomness as in sampling-based methods i.e. every run gives the same results. In that sense, it is like mini-bucket based methods, where results are the same if the variable elimination order and partitioning technique used to generate the initial choice of mini-buckets are the same.
>
> To answer your question, we introduced a random shuffle before finding the disjoint cliques/subsets of factors added. Table 2 in the pdf attached with the rebuttal has the results for two runs. It can be seen that the difference in average $HD_{avg}$ is less than 0.08 and that in average $HD_{max}$ is less than $0.25$ for all benchmark sets. Maximum variation is seen in Grids and DBN where all initial factors have the same size and no reduction in the number of factors is possible since the size of the maximal cliques is also two.
>
> We can include this discussion in the revised version. But, unless we know the variability in other methods, it will not be a fair comparison.
>
> >  contribution appears to be quite incremental over previous work on IBIA framework.
>
>  Please refer to the global rebuttal.
>
> > I think a deeper discussion on the connection between previous schemes and the proposed IBIA based one is clearly called for
>
>  Thank you for pointing out Frank Hutter's work on incremental thin JTs. We did not know of this work, mainly because it seems to be unpublished and the authors did not pursue it further.  To understand it,  we hand-worked a few examples and the main problems are summarized in the global rebuttal.
>
> We hope this addresses all your concerns.

---

### Official Review · Reviewer_SJmy · 2023-07-06

**Soundness:** 3 good
**Presentation:** 2 fair
**Contribution:** 2 fair
**Rating:** 6
**Confidence:** 4

**Summary:**

The paper proposes a new approximate inference algorithm for probabilistic graphical models (PGMs). The approach uses a framework called IBIA to perform marginal inference in discrete PGMs. The main idea is to perform Belief Propagation on a sequence of clique trees. In each step the clique tree is approximated such that the number of variables in any factor is less than a bound and BP is performed on this approximate tree. Factors are then added to this tree and the messages are updated using heuristics to get the next clique tree in the sequence.

**Strengths:**

The approach proposed in the paper seems to work well over a large set of UAI benchmarks in comparison with well-known inference methods.
The idea of computing marginals based on a sequence of linked Clique Tree Forests seems novel.
The idea of running BP for a single iteration to calibrate and obtaining good experimental results seems interesting since it can reduce computational complexity.

**Weaknesses:**

The presentation could be improved since it seems a bit hard to understand all the steps. Perhaps a pseudo-code like algorithm might have helped. Also, since there have been a lot of approximate inference algorithms in PGMs, the main motivation for this approach was a bit unclear. For example, if I understand correctly the mini-bucket methods (Dechter 2003) also have a similar idea of approximating factors through mini-buckets. I think in the related work section the significance of the method can be perhaps better motivated in relation to these methods.

One aspect that was not very clear was when BP converges on the approximate clique tree, if we change the cliques (by adding factors) we cannot just update the beliefs without performing inference again I assumed. In that case, if we use heuristics to update the beliefs (as is mentioned) won’t the errors propagate through the sequence of clique trees? I think there are convergence guarantees on he marginals (e.g. as we add more clique trees in the sequence), that seems like a significant drawback since the scheme will not be any-time (as compared to several other techniques which have this property).

**Questions:**

What is the effect of longer vs shorter sequences, how will this affect the experiment results?
How is the idea conceptually related to other methods such as mini-buckets, etc.?

**Limitations:**

Limitations are mentioned.

---

> ### Author Rebuttal · Authors · 2023-08-09
>
> Thank you for your comments and questions. Following are our responses to each question.
>
>  > What is the effect of longer vs shorter sequences, how will this affect the experiment results?
>
>  Error is introduced both in the approximate step during sequence construction and also during heuristic belief update in the backward pass. Since both these steps are performed for each CTF in the sequence, errors for longer sequences are expected to be larger in general. Increasing the maximum clique size bound $mcs_p$ allows for the addition of more factors to a single CTF and thus, leads to shorter sequences and smaller errors in general.
>
> This is illustrated in Table 1 of the paper (page 8) which shows that IBIA with $mcs_p=23$ gives smaller errors than $mcs_p=20$. We have tabulated the corresponding average sequence length for both these cases in Table 3 in the pdf attached with this rebuttal. As seen from the table, on average, sequence length reduces with an increase in $mcs_p$.
>
> > How is the idea conceptually related to other methods such as mini-buckets, etc.?
>
> Please refer to the global rebuttal.
>
> > The presentation could be improved since it seems a bit hard to understand all the steps. Perhaps a pseudo-code like algorithm might have helped.
>
> We have included the pseudo-code in the supplementary material (Page 3).
>
> > One aspect that was not very clear was when BP converges on the approximate clique tree, if we change the cliques (by adding factors) we cannot just update the beliefs without performing inference again I assumed.
>
>  That is correct. After adding new factors, we need to perform inference again. This is in the forward pass while constructing the sequence of CTF.
>
> > In that case, if we use heuristics to update the beliefs (as is mentioned) won’t the errors propagate through the sequence of clique trees?
>
> Yes. However, our approximation method and the heuristics used for belief update maintain the accuracy. This is demonstrated by our empirical evaluation.
>
> > I think there are convergence guarantees on he marginals (e.g. as we add more clique trees in the sequence), that seems like a significant drawback since the scheme will not be any-time (as compared to several other techniques which have this property).
>
> Similar to mini-bucket based approaches, IBIA can be run in an anytime manner by changing the maximum clique size bound $mcs_p$. This is illustrated in Table~1 (Page 8). For time limit of 2 min, solutions are obtained with $mcs_p$ set to 20. For time limit of 20 min, solutions with improved accuracies could be obtained by increasing $mcs_p$ to 23.
>
> We hope this addresses all your concerns.

---

### Official Review · Reviewer_vgXS · 2023-07-06

**Soundness:** 3 good
**Presentation:** 3 good
**Contribution:** 3 good
**Rating:** 7
**Confidence:** 3

**Summary:**

The paper proposes a new approximate inference method for marginals of probabilistic graphical models (PGMs) based on the incremental build-infer-approximate (IBIA) framework. IBIA was introduced by Bathla & Vasudevan in an arXiv preprint in April 2023 to estimate the partition function (normalizing constant). It achieves this by converting the PGM into a sequence of clique tree forests (SCTFs) with a user-defined clique size bound. The sequence is constructed iteratively: the first step (Build) adds factors of the PGM to the current CTF until the maximum clique size is reached; the second step (Infer) calibrates all clique trees using standard belief propagation; and the final step (Approximate) reduces the clique size to another user-defined bound.

This paper builds upon the previous IBIA work in the following ways: it observes and proves several properties of the SCTF generated by the IBIA framework, which it uses to design an extension of IBIA to approximate marginals (in addition to the partition function). This extension works by introducing links between cliques in subsequent steps of the SCTF based on shared variables, called sequence of linked CTFs (SLCTF). It backpropagates beliefs from the last CTF to previous CTFs via the links and re-calibrates each CTF with one round of message passing. Finally the marginal distribution can be inferred from the CTF containing the desired variable. The paper also describes heuristics for the choice of links and notes that in the directed case (i.e. Bayesian networks), it is advantageous to add variables in topological order during the incremental build step.

The experimental evaluation compares with two existing methods, loopy belief propagation (LBP) and iterative join graph propagation (IJGP), on benchmarks from recent UAI competitions with time limits of 2 min, 20 min, and, for some benchmarks, 1h. They report the number of instances solved within the time limit and the average and maximum Hellinger distance over all non-evidence variables. The new method performs performs better than LBP and IJGP on most examples and is similar on the remaining ones, even though IBIA is implemented in Python whereas the baselines are written in C++.

**Strengths:**

The proposed approximate inference method is interesting and improves the performance compared to existing methods (LBP and IJGP). The extension of the IBIA framework with links is new and useful. The explanation is adequate given the page constraints and the use of a running example is very helpful to understand the algorithm.

**Weaknesses:**

The contribution of this paper (adding links) seems to me to be relatively minor compared to the introduction of IBIA itself in previous work. Extending it in some way to relate successive CTFs (here done via links) seems quite obvious (although that might be the benefit of hindsight).

Regarding the experimental evaluation, I had a look at the recent UAI 2022 competition, since the paper mentions that the same benchmarks were used for recent UAI competitions, if I understand line 14 correctly. It looks like IBIA was a competitor there and while performing well, it did not come in first, as the experimental evaluation of this paper might suggest. For this reason, I'm not convinced that the baselines in the experimental section are representative of the field of PGM inference and whether comparisons with more methods are needed to get an accurate picture of IBIA's performance (see my question to the authors below).

As a minor point: while the presentation is generally fine, I found the notation confusing at times, see below.

**Questions:**

(I will update my rating if the author's answers address my concerns.)

My main question is about the UAI 2022 competition: it seems to me that other submissions there performed better than the baselines in this paper (LBP, IJGP). Is this understanding correct? If so, what is the reason for choosing what seem to be inferior baselines?

I was also confused by line 375, which seems to say that topological ordering gives large errors sometimes. Previously, it sounded like topological ordering was essential for good performance and for the theoretical results in Section 3.1 (line 163). Can you clarify the drawbacks and benefits of topological ordering? Do you know why it causes large errors sometimes?

I also noticed some confusing notation, which I mention here for lack of a better place in the review form:
- equation (1): (i - j) looks like subtraction. In graph theory, an edge is often written as a pair (i, j) instead.
- equation (3): I don't understand the point of the quantifier ($\forall$) in the product notation and have never seen this before.
- line 100: $CTF_{k-1,a}$ first looked like a double-index to me, but I assume the "a" just stands for "approximation".

Typos:
- line 69: I think it should be $Scope(\phi_\alpha)$
- Figure 1, incremental build: I think it should be "while" instead of "until"

**Limitations:**

The authors have adequately discussed the limitations of their work.

---

> ### Author Rebuttal · Authors · 2023-08-09
>
> Thank you for your comments and questions. We would also like to thank you for pointing out the typos. We will fix them in the revised version.  Following are our responses to each question.
>
>
> > My main question is about the UAI 2022 competition: it seems to me that other submissions there performed better than the baselines in this paper (LBP, IJGP). Is this understanding correct? If so, what is the reason for choosing what seem to be inferior baselines?
>
> Apart from IBIA, there were two other solvers for the MAR task in UAI2022 competition.
> 1. $uai14$\_$mar$: While this solver had the highest score, it is an amalgam of solvers that dumps solutions with different methods based on the available time and memory constraints. As described by the participants, "they begin by reparameterizing the model using loopy BP, then build a series of generalized BP reparameterizations whose outer regions are selected via mini-bucket, eventually transitioning, when there is much more time than memory to cutset conditioning of GBP approximations."
>
> * The implementation of $uai14$\_$mar$ is not publicly available. Therefore, we have individually compared our results with methods that belong to categories of methods used in $uai14$\_$mar$, namely, Loopy BP (LBP), IJGP which performs generalized BP on loopy graphs obtained using mini-bucket heuristics and IJGP with cutset sampling (ISSwc).
> For IJGP and ISSwc, we have used open-source implementations by the authors and for LBP, we have used the LibDAI tool which has been used for comparison in several latest works (Shih, A. \& Ermon, S., 2020; Agrawal, D.K., Pote, Y., \& Meel, K.S. (2021)) and was among the winners of UAI2010 competition. We want to point out that as a standalone method, IBIA performs better than each individual solver used in the amalgam.
>
>
> 2. $lbp$\_$mar$: This performs loopy BP and was seen to give much lower scores than IBIA in the 20 min category, which is consistent with our results.
>
> We would also like to point out that we have compared the average Hellinger distances over all instances in each benchmark set (total of 493 instances). However, in the competition, the reported score (which is not directly Hellinger distance)  is the average score over a subset of 88 instances from different benchmarks in the final set and 123 instances in the tuning set. IBIA had a comparable score as $uai14$\_$mar$ for the tuning set and lower for the final set. While the overall trend can be compared, a direct one-to-one comparison should not be made.
>
> > Can you clarify the drawbacks and benefits of topological ordering? Do you know why it causes large errors sometimes?
>
> The advantage of topological ordering is that once all the evidence variables are added,  no belief-update is needed for the subsequent CTFs (using theoretical results discussed in Section 3). So, the number of belief-update steps is lower, resulting in lower errors.
>
> The drawback of topological ordering is that it is rigid and it sometimes results in a larger number of CTFs in the sequence which could give rise to larger errors if all the evidence variables are added in later CTFs.
>
> We can add this to the discussion.
>
> > I also noticed some confusing notation, which I mention here for lack of a better place in the review form:
> >* equation (1): (i - j) looks like subtraction. In graph theory, an edge is often written as a pair (i, j) instead.
> >* equation (3): I don't understand the point of the quantifier ($\forall$) in the product notation and have never seen this before.
> >* line 100: $CTF_{k-1,a}$ first looked like a double-index to me, but I assume the "a" just stands for "approximation".
>
> We will fix Equations (1), (3) in the revised version.
>
> Yes, "a" indeed stands for approximation. $CTF_{k-1,a}$ is the approximate CTF obtained after approximating $CTF_{k-1}$.
>
>
>
> We hope this addresses all your concerns.

---

> > ### Comment · Reviewer_vgXS · 2023-08-15
> >
> > Thank you for providing such an extensive response. My concerns regarding the baselines and the UAI2022 competition have been fully addressed and I don't see a problem with the evaluation anymore. Please add your response to the discussion of the topological ordering in the paper; it clarifies a lot. I have raised my rating from 5 to 7 as a result of the rebuttal.

---

> > > ### Author Response · Authors · 2023-08-17
> > >
> > > Thank you for your prompt response. We will certainly add the discussion on topological ordering to the revised version.

---

### Author Rebuttal · Authors · 2023-08-09

We thank all reviewers for their insightful comments and questions. We are encouraged they found our idea to be novel (**vgXS**, **SJmy**), useful (**vgXS**) and interesting (**vgXS**, **SJmy**).  We are glad that the reviewers found our evaluation sound (**QKYy**) and that our method is competitive (**QKYy**) and improves the performance compared to existing methods (**vgXS**, **SJmy**,**QKYy**). We are also pleased that the reviewers found our paper well-written, organized and self-contained with adequate explanations (**QKYy**, **vgXS**). There seem to be two primary concerns, first regarding the contributions of this work and the second regarding comparison with related work. We would like to address these here. Detailed responses to other questions asked by reviewers are included in individual rebuttals.


[**vgXS**, **SJmy**, **QKYy**]  *Regarding the contribution of this paper:*

Although there is a plethora of approximate inference methods in the literature, most of these have been evaluated for the task of finding partition function (PR).  Only a few of these methods have published results for finding posterior marginals (MAR) of all variables which is a key inference task in several applications.  Even in the UAI 2022 competition, while many solvers were submitted for the PR, MPE, MMAP tasks, only a few were submitted for the MAR task.  Publicly available implementations for other submitted solvers like  wmbsearch,  abstraction sampling, toulbar, daoopt and braobb do not support the MAR task. There seems to be no obvious way to extend them for the MAR task. In this context, showing that IBIA can be extended for MAR is a significant contribution.

There is nothing in the base paper (Bathla \& Vasudevan, 2023) to suggest an obvious way to find MAR. It took us a while to figure out that these links can be added and used for belief update. Coming up with reliable algorithms that give good accuracies, understanding differences between topological and non-topological ordering as well as studying the theoretical guarantees for topological ordering is a non-trivial extension.

[**SJmy**, **QKYy**] *Regarding related work:*

IBIA is similar to mini-bucket schemes in the sense that the accuracy-complexity tradeoff is controlled using a user-defined maximum clique size bound. While mini-bucket based schemes like iterative join graph propagation (IJGP) and join graph linear programming (JGLP) use iterative message passing in loopy cluster graphs,  others like mini-bucket elimination (MBE), weighted MBE (WMB) and mini-clustering are non-iterative approaches that approximate the messages by migrating the sum operator inside the product term. In contrast,  IBIA constructs a sequence of clique *trees*. It performs belief propagation on approximate clique *trees* so that messages are exact and there are no issues of convergence.

In order to construct the sequence, IBIA requires a fast and accurate method to approximate clique trees by reducing clique sizes.
The aim is to preserve as much as possible, the joint distribution of the subset of variables present in factors that have not yet been added to a clique tree (called interface variables in this work). This is accomplished by marginalizing out variables from large-sized cliques and a minimal set of neighbors without disconnecting clique trees. IBIA gives good accuracy since variables that are removed are chosen based on a mutual information (MI) based metric and a sufficient number of non-interface variables are retained so that a CT is never disconnected. In contrast, the Boyen Koller (BK) (Boyen, 1998) and factored frontier (FF) approximations, although fast, retain only the interface variables which can disconnect the CT resulting in larger errors due to the underlying independence approximation.  The thin junction tree approximations proposed in Frank Hutter et. al (2004) and Kjærulff (1994) split cliques that contain a pair of variables that is not present in any of its sepsets. However, if there are large cliques that have only sepset variables (which is typical in most benchmarks), then the split has to be done iteratively starting from leaf nodes of multiple branches, until the large-sized clique can be split. When such cliques are centrally located in the CT, this process is both time-consuming and would result in approximation of a much larger set of cliques. Similar to BK and FF, this method can also disconnect the CT.

The other thin junction tree methods choose an optimum set of features based on either KL distance or a MI based metric. They cannot be directly used since IBIA requires a good approximation of the joint distribution of only the interface variables. Also, these methods are typically iterative and not very fast.

 We can include this in our comparison with related work.

---

### Decision · Program_Chairs · 2023-09-21

**Decision:**

Accept (poster)

**Comment:**

This paper proposes a novel approximation scheme for discrete graphical model marginal inference that proves to be competitive on the standard UAI competition benchmarks. The authors responded very clearly to the questions raised by the reviewers.